# Final-Term Report: Learn To Manage Portfolio With Reinforcement Learning

**Yuan LIU**
Department of Engineering
The Chinese University of Hong Kong
Shatin, Hong Kong
1155148271@cuhk.edu.hk

**Qianyu ZHOU**
Department of Engineering
The Chinese University of Hong Kong
Shatin, Hong Kong
503105037@qq.com

## 1   Introduction

Utilizing deep reinforcement learning in portfolio management is gaining popularity in the area of algorithmic trading. Our group find this subject is closer to our major, so we try to find some use of deep reinforcement learning in this area. And finally, we choose to make a study about if deep reinforcement learning tools can really help us manage people's investment portfolios and if the agent can actually beat fund managers in some circumstance. In our experiment, we use two deep reinforcement learning algorithms: Policy Gradient (PG) and Proximal Policy Optimization(PPO) to explore the impact of different optimizers and network structures on the training of trading agents. We utilize a data set based on the American stock market, finally reach our own experimental conclusions and improvement plans.

## 2   Environment and Data

The source of our experimental data is Wind. We select low-correlation or even negative-correlation stock sets in the markets to show our agent's ability to allocate different assets. In order to maintain our assumptions, we choose stocks with large trading volumes to ensure that our actions will not affect the market. In the American stock market, we select 5 stocks to test the performance of our agents on large-scale asset allocation issues. Their stock codes are "BABA", "AAPL", "V", "SNE", "ADBE" separately. In addition, we choose the past three years as our training and testing period. Data from 2015, January 5th to 2017, June 27th is our training data and data from 2017, June 28th to 2017, December 29th is our testing data. In order to derive a general agent which is robust with different stocks, we normalize the price data. To be specific, we divide the opening price, closing price, high price and low price by the close price at the last day of the period. For missing data which occurs during weekends and holidays, in order to maintain the time series consistency, we fill the empty price data with the close price on the previous day and we also set volume 0 to indicate the market is closed at that day.

## 3   Model and Algorithm

At present, in terms of asset management problem, a series of different reinforcement learning methods have been proposed by the academic circle and the industry, such as PG, dual Q network, DDPG, PPO, etc. And in our group's experiment, we will use the PG algorithm and the PPO algorithm for experiments here to test the potential of the algorithm in asset allocation.

Most algorithms for policy optimization can be classified into three broad categories: policy iteration methods, policy gradient methods and derivative-free optimization methods. Policy Gradient(PG) is one of them and is commonly used in training agent to deal with financial product investment. So our group determines to use this algorithm first to execute our project.

---

**Algorithm 1** Policy Gradient for stock Trading

---

1: Randomly Initialize actor $\mu\left(s \mid \theta^\mu\right)$
2: Initialize replay buffer R
3: **for** i = 1 to M **do**
4:     Receive initial observation state $s_1$
5:     Add noise into the price data
6:     **for** t = 1 to T **do**
7:         Select action $w_t = \mu\left(s_t \mid \theta^\mu\right)$
8:         Execute action $w_t$ and observe $r_t, s_{t+1}$ and $w'_t$
9:         Save transition $(s_t, w_t, w'_t)$ in R
    end for
10:    Update actor policy by policy gradient:
11:    $\bigtriangledown_{\theta^\mu} J = \bigtriangledown_{\theta^\mu} \frac{1}{N} \sum_{t=1}^{T} \log(w_t \cdot y_t - \mu \sum_{i=1}^{m} \mid w_{i,t} - w'_{i,t-1} \mid)$
    end for

---

Proximal Policy Optimization(PPO) is also one of the policy gradient methods. PPO is based on Trust Region Policy Optimization(TRPO), TRPO finds a lower bound for policy improvement so that policy optimization can deal with surrogate objective function, this could guarantee monotone improvement in policies. And PPO proposes new surrogate objective to simplify TRPO.

---

**Algorithm 2** Proximal Policy Optimization for stock Trading

---

1: Initialize actor $\mu : S \rightarrow \mathbb{R}^{m+1} and$
2: $\sigma : S \rightarrow diag\left(\sigma_1, \sigma_2, \cdots, \sigma_{m+1}\right)$
3: **for** i = 1 to M **do**
4:     Run policy $\pi_\theta \sim N(\mu(s), \sigma(s))$ for T timesteps and
5:     collect $(s_t, a_t, r_t)$
6:     Estimate advantages $\hat{A}_t = \sum_{t'>t} \gamma^{t'-t} r_{t'} - V\left(s_t\right)$
7:     Update old policy $\pi_{old} \leftarrow \pi_\theta$
8:     **for** i = 1 to N **do**
9:         Update actor policy by policy gradient:
10:        $\sum_i \nabla_\theta L_i^{CLIP}(\theta)$
11:        Update critic by:
12:        $\nabla L(\phi) = -\sum_{t=1}^{T} \nabla \hat{A}_t^2$
        end for
    end for

---

# 4   Result and Prediction

After our training and testing, our group has made some progress. As you can see in the chart as below, we use four agent to train our data, Winner, Loser, UCRP, PG, the first three algorithm is the traditional way to trade the portfolio, the final one we try to reimplement the PG algorithm, and the result shows as below:

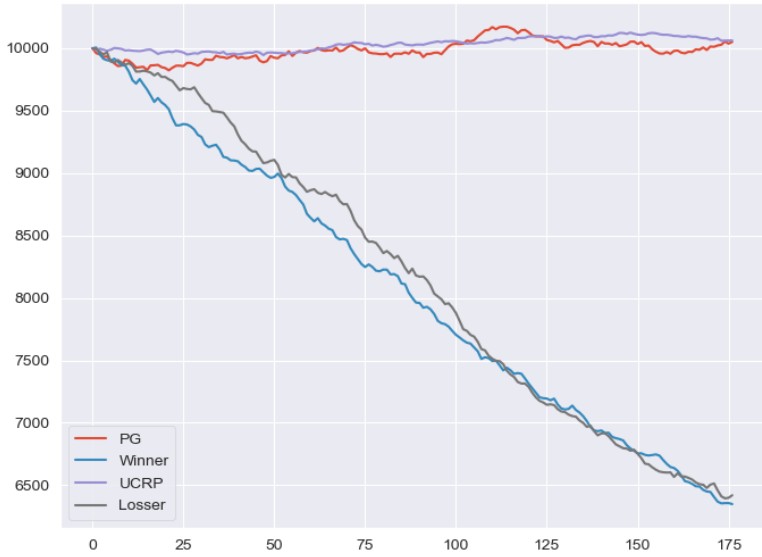

Figure 1: Result of 4 Agent, 100 epochs

The red line shows how our PG agent acts after the training period of 100 epochs. Basically this line is above the initial value of 10000 at most of the time, and shows a little bit more volatility compared with the traditional UCRP method. This graph shows that our agent can at least beat the Winner and the Loser's strategy, and be able to perform as good as the UCRP strategy.

As we print the ARR and Sharpe Ratio of the agent as below:

| asset | ARR | Sharp Ratio | Backtest |
|---|---|---|---|
| PG | 0.602 % | 1.223 | 0.023 |
| Winner | 0.351 % | 0.711 | 0.349 |
| UCRP | 0.607 % | 1.233 | 0.015 |
| Losser | 0.344 % | 0.698 | 0.358 |

Figure 2: ARR of the Agent

We can find that the PG strategy shows nearly the same ARR and Sharpe Ratio with the UCRP strategy, which means their expected value of the excess of the asset return and standard deviation of the asset excess return are very close. However, the Winner's strategy and the Loser's strategy don't show a good return compared with the first two strategies.

After our training of 100 epochs, we want to find out if our agent can perform better if we enhance the epochs of training. And we find the overfitting phenomenon when we train this agent with 1000 epochs. It shows a dramatic bad performance in the trading result. And we find the suitable epoch is approximately from 50 to 100 when the agent acts best during the trading process. Figure 3 and figure 4 demonstrate the performance of our agent separately in different epochs of training.

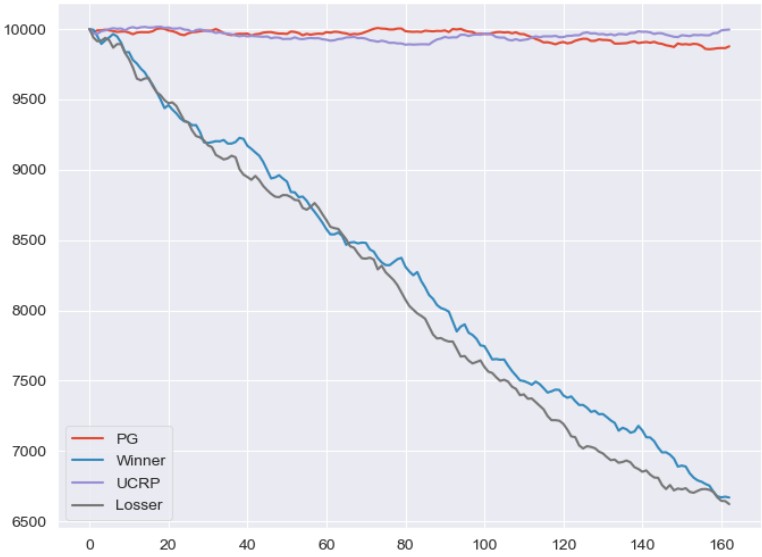

Figure 3: Result of 4 Agent, 200 epochs

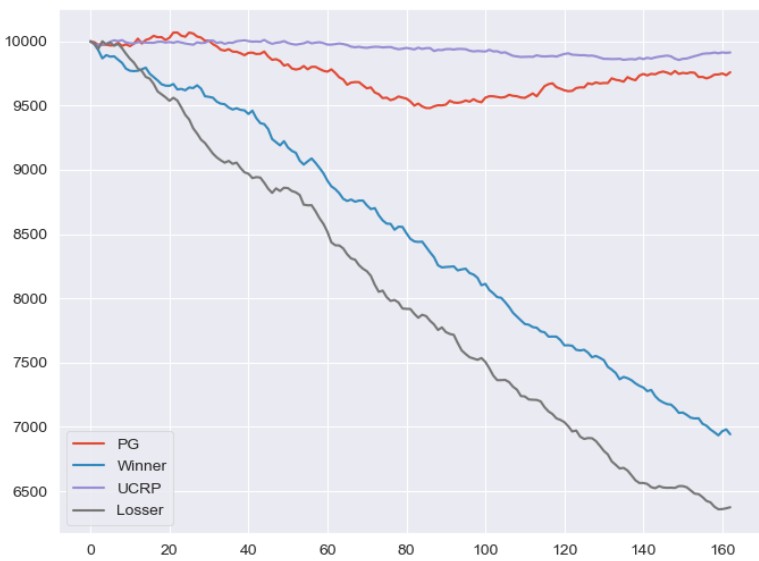

Figure 4: Result of 4 Agent, 1000 epochs

Furthermore, as we already know the performance of PG training agent, we want to investigate if PPO training agent can perform better and can beat previous one. But the result basically shows that PPO perform much worse than PG. We train our agent with PPO and the result is as below:

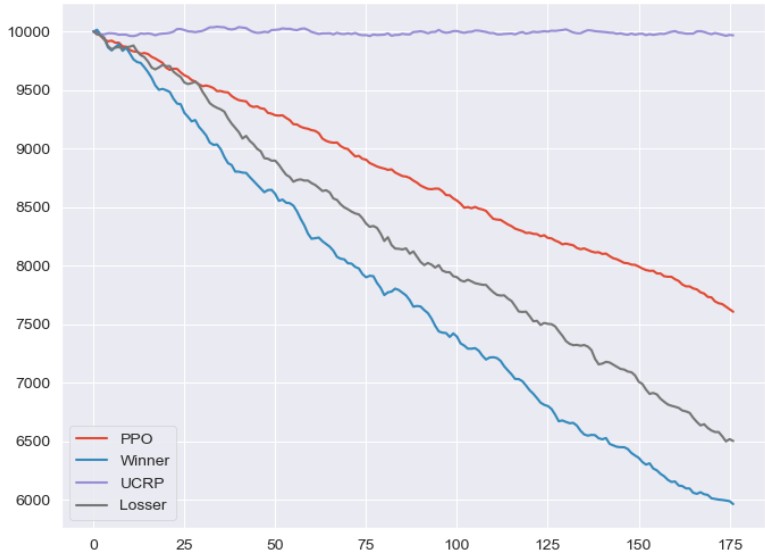

Figure 5: Result of 4 Agent, 100 epochs

And the ARR and Sharpe Ratio of the agent are shown as below:

| asset | ARR | Sharp | Backtest |
|---|---|---|---|
| PPO | 0.411 % | 0.868 | 0.239 |
| Winner | 0.274 % | 0.577 | 0.405 |
| UCRP | 0.563 % | 1.193 | 0.008 |
| Losser | 0.322 % | 0.681 | 0.35 |

Figure 6: ARR of the Agent

As the graph demonstrates, PPO agent performs much worse than PG in this financial product investment process. It shows a dramatic drop with its principal after half a year's investment behavior. Its Sharpe Ratio only equals to 0.868, compared with the number 1.223 of the PG agent, this result means its expected value of the excess of the asset return divided by standard deviation of the asset excess return is relatively low, and this invest behavior doesn't seem to be acceptable.

We then try to enhance or reduce the epochs of training period, in order to find if we overfit the data or on the contrary. But we regretfully found that this doesn't lead to a better performance for PPO agent, no matter how we change the number of epochs, the curve just goes downwards. So in our circumstance, we have drawn the conclusion that PG is more suitable than PPO method to train the agent to make a deal with American stocks.

