# OpenReview forum: "Learn To Manage Portfolio With Reinforcement Learning"
_CUHK.edu.hk/2021/Course/IERG5350_

### Official Review · AnonReviewer3 · 2020-12-15
**review feedback**

**Rating:** 7
**Confidence:** 4

**Review:**

Quality:  met the acceptable quality bar as an experimentation project.

Clarity: The structure of this paper is clear; but it doesn't capture the details of how the Portfolio Management is turned into RL problem. It's good to  provide definitions of States, Actions, Rewards at least; optionally Environment as well if it's done in any stock backtesting environment.

Originality: relatively lack of originality given the scope is limited only on implementing existing RL algo for portfolio management.
Significance: relatively lack of significance given the scope.

Pros: explaing well about test data and performance matics such as Sharpe Ratio.
Cons: 1)as mentioned in 'Clarity', it will be greate to explain those missing details; 2) conclusion about what exactly can be further improved or experimented.

---

### Official Review · AnonReviewer1 · 2020-12-20
**A case study of RL algorithm in portfolio management**

**Rating:** 4
**Confidence:** 5

**Review:**

This paper make a case study of RL algorithm in portfolio management. Author claimed that they have made some improvement for RL algorithm in portfolio management field. Exploring the impact of different optimizers and network structures on the training of trading agents.

I think this paper need to make a big adjustment. There are not any substantive improvement for the RL algorithm. What I find is that author only modified the training time of the PPO. The paper is hard to follow there are too much terms and variables not introduced. The RL algorithm in the paper is not clear, since the prescribed issue. Overall I do not think this is a qualified paper for the final project. If author can sovle the issues I proposed I will increase my score.



### Pros:

* Conducting some experiments to show the performance of RL algorithm.
* Reproduceing some algorithms in portfolio management.

### Cons:
* I think the speaking speed  in video representation is too fast and been modified. So it is hard to follow.

* The Structure of paper is not well orgnized, the abstract, related works, conclusion are missed. The pictures in the paper is too large. I think author can group the experimental results into one page.

* The way of displaying the experiment result is hard to understand. Why the chart of comparison of four algorithms only exhibit the results after 100 epochs? If so why the label of axis is still start from 0. The label of y axis is stand for what?

* What are meaning of the evaluation metrics you have shown in the figure2 and figure6, the evaluation metric should be introduced before you show the results.


### Some suggestion:

* More detail of the experiment, algorithm and related works. Since this paper is focus on the case study a throughly analysis, background introduction and experiment set up are needed.

* More insight of how to improve or adapt the PPO into the portfolio management field.

---

### Official Review · AnonReviewer2 · 2020-12-20
**Manage Portfolio with policy-based methods**

**Rating:** 6
**Confidence:** 4

**Review:**

General:
The paper tried to explore the application of reinforcement learning on portfolio management. To maximize the reward, they used some policy-based methods to train their agent and proved it works to help their agent to get better performances on the financial product investment process.

Evaluation of the quality: This paper proposed to use a policy-based method for portfolio management. After the training, their agent performs better than the methods with Winner and Losser strategy.

Clarity:
The paper is well-organized. However, there are some questions. Firstly, there is no conclusion section in this paper. Then, more illustrations about the figures are needed in section 4. For instance, what the meaning of the number in the y-axis. What’s more, as a paper of reinforcement learning, the paper needs to clearly explain their states, actions, and reward, etc.

Originality:
This paper tried PG and PPO algorithms. Regrettably, this paper doesn’t mention the details of the two algorithms they used in this model, for example, the network structure they used in the PPO algorithm.

Significance:
Learn To Manage Portfolio With Reinforcement Learning is a field with great potential. This paper tried to use PPO and PG to train their model. Though there are some similar papers and works in this area, I think it meets the requirements of the course project.

Pros:
1.	This paper is well-organized.
2.	The table and graphs show the performance of the proposed approach is good.

Cons:
1.	This paper doesn’t have a clear problem formulation, they need clearly explain their states, actions, and reward, etc.
2.	They need more illustrations about their figures.
3.	More analysis of the results is needed, for instance, why PPO behaves badly.

(By the way, the video sound at 1x speed is a bit strange)